# Image Segmentation Using Active Contours with Hessian-Based Gradient Vector Flow External Force

**DOI:** 10.3390/s22134956

**Published:** 2022-06-30

**Authors:** Qianqian Qian, Ke Cheng, Wei Qian, Qingchang Deng, Yuanquan Wang

**Affiliations:** 1School of Computer Science, Jiangsu University of Science and Technology, Zhenjiang 212003, China; 202070044@stu.just.edu.cn (Q.Q.); 199070046@stu.just.edu.cn (Q.D.); 2School of Electronics and Information, Jiangsu University of Science and Technology, Zhenjiang 212003, China; 211110303118@stu.just.edu.cn; 3School of Artificial Intelligence, Hebei University of Technology, Tianjin 300401, China

**Keywords:** gradient vector flow, Hessian matrix, image structure, anisotropy

## Abstract

The gradient vector flow (GVF) model has been widely used in the field of computer image segmentation. In order to achieve better results in image processing, there are many research papers based on the GVF model. However, few models include image structure. In this paper, the smoothness constraint formula of the GVF model is re-expressed in matrix form, and the image knot represented by the Hessian matrix is included in the GVF model. Through the processing of this process, the relevant diffusion partial differential equation has anisotropy. The GVF model based on the Hessian matrix (HBGVF) has many advantages over other relevant GVF methods, such as accurate convergence to various concave surfaces, excellent weak edge retention ability, and so on. The following will prove the advantages of our proposed model through theoretical analysis and various comparative experiments.

## 1. Introduction

Image segmentation is a key step from image processing to image analysis. Traditional segmentation methods include threshold [1], clustering [2], active contour model [3], region growth [4], etc. Since someone proposed the snake or active contour model in 1988, the snake or active contour model has become one of the mainstream models of image segmentation [3]. Generally, an active contour performs image segmentation by minimizing the combination of internal and external energy and deforming the curve on the image plane; the internal energy keeps the curve continuous and smooth, while the external energy attracts the curve to the boundary of the object to be segmented on the image. Therefore, the problem of finding the boundary of the segmented object can be transformed into the problem of minimizing the internal and external energy. According to the representation of the curve, the active contour is divided into a parametric contour and geometric contour. The parametric model uses explicit parameter representation [3,5,6,7,8,9], and uses image edge mapping to stop the evolution of contour. Parametric models rely heavily on high gradient amplitudes to extract object boundaries, and are effective only when the contrast between background and foreground is clear enough. The geometric model [10,11,12,13,14,15,16,17,18,19,20,21,22,23,24] is based on the theory of level set technology and usually adopts specific regional homogeneity criteria to guide the evolution of contour.

External force plays a leading role in the evolution of parametric snake contour, so people have invested a lot of energy in the research of external force to improve the robustness of active contour. At present, the proposed gradient vector flow (GVF) [25] is still one of the most successful methods. It spreads the gradient vector from the object boundary to the rest of the image, which not only expands the capture range, but also weakens the influence of noise to a certain extent. Due to its effectiveness, a large number of fast algorithms for the GVF model have been proposed, including vector field convolution (VFC) [26], BVF [27], GVF based on augmented Lagrange [28], the multi-grid method of GVF [29], and efficient numerical format of GVF [30]. Some other efforts focus on improving the initial edge map, for example, a guided filter is employed to enhance the initial edge map [31,32] and a directional edge map is coined for the GVF model [33]; in the literature, the GVF is modified by using the initial contour position and introducing additional boundary conditions of Dirichlet type [34]. Many efforts pay attention to reformulating the energy functional of the GVF model, among others, examples include the harmonic gradient vector flow (HGVF) [35], harmonic surface [32,36], 4DGVF external force field [37], NGVF [38], EPGVF [39], MGVF [40], and CN-GGVF [41]. Recently, the GVF model also has some interesting applications, as well as some interesting work on GVF snake initialization for ultrasonic image segmentation, such as walking particles [42,43]. Very recently, Jaouen proposed an image enhancement vector field based on the partial differential equation (PDE) [44], and pointed out the similarity between the vector field and gradient vector flow, which allows a natural connection between impulse filtering and a large number of work on GVF like fields. It is important to note that the deep learning method plays a very important role for image-based applications presently, such as image segmentation [45,46,47,48,49], detection [50,51], and classification [52,53], and it needs big data for training and the active contour is still of importance for image segmentation.

We can see that although the above contents provide various methods to improve the GVF model, they do not consider the characteristics of image structure. Ref. [54] pointed out that the “Hessian method is a method to extract the direction of image features through high-order differentiation”. Inspired by this principle, we express the smooth constraint formula in the GVF model in matrix form, then incorporate the Hessian matrix into the energy functional of the GVF model, and finally get the GVF based on the Hessian matrix, that is HBGVF. Compared with other methods, we experimentally prove that HBGVF has many advantages, such as accurately converging to various concave surfaces while maintaining weak edges. There is more information related to this work in the literature [55,56].

The rest of this paper is arranged as follows: in the next section, we briefly review the snake model and four famous GVF-based external forces, including GVF [25], GGVF [57], VEF [58], NGVF [38], and CN-GVF [41], and compare them with these GVF-based methods through experiments. Section 3 details the HBGVF model proposed in this paper. In Section 4, we prove the advantages of the proposed model through a large number of experiments, and finally draw a conclusion in Section 5.

## 2. Backgrounds

### 2.1. Traditional Model: Active Contours

When the early active contour was proposed, it was defined as the elastic curve c(s)=[x(s),y(s)],s∈[0,1] and the following is its energy function formula: (1)$$E_{\mathrm{snake}\;}=\int \frac{1}{2}\left(\alpha {\left|c^{\prime }\right|}^{2}+\beta {\left|c^{\prime \prime }\right|}^{2}\right)+E_{ext}\left(c\left(s\right)\right)ds$$
in Formula (Equation 1), $c^{\prime }\left(s\right)$, $c^{\prime \prime }\left(s\right)$ are the first and second derivatives of c(s), which are, respectively, positively weighted by α and β. $E_{ext\;}\left(c\left(s\right)\right)$ is the image potential, which may be caused by various things, such as edges. The Euler equation for minimizing $E_{\mathrm{snake}\;}$ can be obtained by deformation calculus as follows: (2)$$\alpha c^{\prime \prime }\left(s\right)-\beta c^{\prime \prime }{}^{\prime \prime }\left(s\right)-\nabla E_{ext}=0$$
Formula (Equation 2) is a force balance equation in reference [8],
(3)$$F_{\mathrm{int}\;}+F_{ext}=0$$
in Formula (Equation 3), $F_{\mathrm{int}\;}=\alpha c^{\prime \prime }\left(s\right)-\beta c^{\prime \prime }{}^{\prime \prime }\left(s\right)$ and $F_{ext\;}=-\nabla E_{ext}$. The internal force $F_{\mathrm{int}\;}$ keeps the snake contour smooth, while the external force $F_{ext\;}$ shrinks the snake contour to the desired image object.

In image I, $F_{ext\;}$ is often used as the gradient vector of image edge mapping, as shown in the following formula: I, as follows,
(4)$$F_{ext}=-\nabla E_{ext}=\nabla {\left|\nabla G_{\sigma }⊗I\right|}^{2}$$

In fact, this gradient vector is local, unable to take into account the overall situation, and is not regular enough, so the snake can not evolve effectively under its guidance.

### 2.2. Gradient Vector Flow (GVF)

Due to the obvious disadvantage of external force in Formula (Equation 4), $F_{ext\;}$ is replaced by a new vector field v=[u(x,y)V(x,y)] in the GVF model, which can be derived by minimizing the following function,
(5)$$E_{GVF}=\;\int \int \;\mu \left(u_{x}^{2}+u_{y}^{2}+v_{x}^{2}+v_{y}^{2}\right)+{\left|\nabla f\right|}^{2}{\left|v-\nabla f\right|}^{2}dxdy$$

In (Equation 5), μ is a positive weight, *f* is the edge map of the image I, and ∇ is the gradient operator. The newly obtained vector field is a gradient vector flow (GVF) field. The GVF field can be obtained by solving the following equation iteratively,
(6)$$\left\{\begin{array}{c} u_{t}=\mu \Delta u-{\left|\nabla f\right|}^{2}\left(u-f_{x}\right)\\ v_{t}=\mu \Delta v-{\left|\nabla f\right|}^{2}\left(v-f_{y}\right) \end{array}\right.$$
where Δ is the Laplacian operator. The diffusion equation is isotropic.

The generalized GVF (GGVF) is an extension of the GVF by replacing μ and $f_{x}^{2}+f_{y}^{2}$ in (Equation 6) with two spatially varying functions $g\left(|\nabla f|\right)=\exp\left(-{\left|\nabla f\right|}^{2}/k^{2}\right)$ and h(|∇f|)=1−g(|∇f|), respectively [57], *k* acts as a threshold and controls the smoothing effect. The introduction of such terms makes the GGVF snake behave better than the GVF snake on thin concavity convergence.

### 2.3. Virtual Electric Field (VEF)

Reference [58] proposed a virtual electric field model (VEF). In this method, each pixel in the image is regarded as an electron, the charge is the size of the image edge, and the virtual electric field at $\left(x_{0},y_{0}\right)$ is derived from the sum of all other electrons in the surrounding area D, which is expressed by the following formula,
(7)$$\begin{array}{l} E_{VEF}\left(x_{0},y_{0}\right)=\\ \sum_{(x,y)\in D}\left(\frac{\left(x_{0}-x\right)}{{\left(\sqrt{{\left(x_{0}-x\right)}^{2}+{\left(y_{0}-y\right)}^{2}}\right)}^{3}},\frac{\left(y_{0}-y\right)}{{\left(\sqrt{{\left(x_{0}-x\right)}^{2}+{\left(y_{0}-y\right)}^{2}}\right)}^{3}}\right)\cdot f\left(x,y\right) \end{array}$$
in Formula (Equation 7), $D=\left\{\left(x,y\right)|-t\leq x_{0}-x\leq t,-t\leq y_{0}-y\leq t\right\}$, *f* is the size of the image edge image. Fast Fourier transform (FFT) is applied to the VEF model, so Formula (Equation 7) is usually written in convolution form, as follows,
(8)$$E_{VEF}\left(x,y\right)=\left(-\frac{x}{{\left(\sqrt{x^{2}+y^{2}}\right)}^{3}},-\frac{y}{{\left(\sqrt{x^{2}+y^{2}}\right)}^{3}}\right)⊗f\left(x,y\right)$$
in Formula (Equation 8), ⊗ represents the convolution operation.

Thanks to the use of FFT, the VEF model can be realized in real time. In addition, the VEF model also has some characteristics better than the GVF model, such as a large capture range and more sensitive concave convergence.

### 2.4. Gradient Vector Flow in Normal Direction (NGVF)

It was pointed out in [59] that the Laplace operator can be decomposed into two terms, as shown below,
(9)$$\Delta u=u_{TT}+u_{NN}$$

Taking u(x,y) as an example, in Formula (Equation 9), $u_{TT}$ and $u_{NN}$ are the second derivatives of u(x,y) in the tangential and normal directions of the isophotes, respectively. It was pointed out in [60] that, as an interpolation operator, $u_{NN}$ has the best performance, Δu second and $u_{TT}$ third. The diffusion process in (Equation 6) is regarded as the interpolation process, and the NGVF is proposed using the optimal interpolator, as shown in the following formula,
(10)$$\left\{\begin{array}{c} u_{t}=\mu u_{NN}-\left(u-f_{x}\right){\left|\nabla f\right|}^{2}\\ v_{t}=\mu v_{NN}-\left(v-f_{y}\right){\left|\nabla f\right|}^{2} \end{array}\right.$$
where μ is also a positive weight as in (Equation 6).

### 2.5. Component-Normalized Generalized Gradient Vector Flow (CN-GGVF)

In the CN-GGVF model, the diffusion equations are modified in the following form,
(11)$$\left\{\begin{array}{c} u_{t}=g\left(|\nabla f|\right)\cdot \left(g\left(|\nabla f|\right)u_{NN}+h\left(|\nabla f|\right)u_{TT}\right)-h\left(|\nabla f|\right)\cdot \left(u-f_{x}\right)\\ v_{t}=g\left(|\nabla f|\right)\cdot \left(g\left(|\nabla f|\right)v_{NN}+h\left(|\nabla f|\right)v_{TT}\right)-h\left(|\nabla f|\right)\cdot \left(v-f_{y}\right) \end{array}\right.$$
where the g(|∇f|) and h(|∇f|) are identical to those in the GGVF model, and $u_{TT}$ and $u_{NN}$ are identical to those in the NGVF model. Based on deep analysis of the behavior of the GGVF model, Qin et al. proposed to normalized the GVF vector in a component-wise manner, such that the CN-GGVF model can converge to a deep and thin notch, the component-normalized (CN) GGVF field reads,
(12)$$u_{CN-GVF}=\mathrm{sign}\left(u\right)=\left\{\begin{array}{cc} 1, & u>0\\ 0, & u=0\\ -1, & u<0 \end{array}\right.$$
(13)$$v_{CN-GVF}=\mathrm{sign}\left(v\right)=\left\{\begin{array}{cc} 1, & v>0\\ 0, & v=0\\ -1, & v<0 \end{array}\right.$$

## 3. The HBGVF Model

### 3.1. Gradient Vector Flow Expressed in Matrix Form

By observing equation, we first reformulate the smoothness constraint in the GVF model into matrix form as follows, $u_{x}^{2}+u_{y}^{2}=\left(\begin{array}{cc} u_{x} & u_{y} \end{array}\right)\left(\begin{array}{c} u_{x}\\ u_{y} \end{array}\right)=\left(\begin{array}{cc} u_{x} & u_{y} \end{array}\right)\left(\begin{array}{cc} 1 & 0\\ 0 & 1 \end{array}\right)\left(\begin{array}{c} u_{x}\\ u_{y} \end{array}\right)$, we first reformulate the smoothness constraint in the GVF model into matrix form as follows,
(14)$$E_{GVF}=\;\int \int \;\mu \left[{\left(\nabla u\right)}^{T}\cdot W\cdot \nabla u+{\left(\nabla v\right)}^{T}\cdot W\cdot \nabla v\right]+{\left|\nabla f\right|}^{2}{\left|v-\nabla f\right|}^{2}dxdy$$
in Equation (Equation 14), W is the identity matrix. It can be seen from the above formula that due to the existence of this identity matrix, it induces the scalar L2 norm, so that the GVF model fails to take into account the image characteristic of image structure. We completely replace all W with matrix D related to the image structure, so we use Hessian matrix to construct, as shown below,
(15)$$E=\;\int \int \;\mu \left[{\left(\nabla u\right)}^{T}\cdot D\cdot \nabla u+{\left(\nabla v\right)}^{T}\cdot D\cdot \nabla v\right]+{\left|\nabla f\right|}^{2}{\left|v-\nabla f\right|}^{2}dxdy$$
where $D=\left(\begin{array}{cc} a & b\\ b & c \end{array}\right)$ is a symmetric and positive semi-definite matrix. The reconstructed model is called the Hessian-based GVF (HBGVF for short). Using the variational method, the HBGVF field can be obtained by solving the following equation, as shown below,
(16)$$\left\{\begin{array}{c} u_{t}=\mu div\left(D\nabla u\right)-{\left|\nabla f\right|}^{2}\left(u-f_{x}\right)=0\\ v_{t}=\mu div\left(D\nabla v\right)-{\left|\nabla f\right|}^{2}\left(v-f_{y}\right)=0 \end{array}\right.$$
in Equation (Equation 16), div is the divergence operator.

### 3.2. Using the Hessian Matrix to Construct Diffusion Matrix

Through the observation of Formula (Equation 16), we can know that its equation is exactly the tensor based diffusion in [61]. The “Hessian method proposed in reference [54] regards the direction of the maximum second-order directional derivative as the direction passing through the image feature, and its vertical direction is regarded as the direction along the image feature.” Inspired by this principle, we use Hessian matrix to reconstruct the diffusion matrix D in Formula (Equation 16). Taking image I as an example, its Hessian matrix is represented by the following formula,
(17)$$H=\left(\begin{array}{cc} I_{xx} & I_{xy}\\ I_{xy} & I_{yy} \end{array}\right)$$
using the derivative in [61], the two eigenvalues of H can be solved by the following formula, expressed by $\lambda_{1}$ and $\lambda_{2}$,
(18)$$\left\{\begin{array}{c} \lambda_{1}=\frac{1}{2}\left[\left(I_{xx}+I_{yy}\right)+\sqrt{{\left(I_{xx}-I_{yy}\right)}^{2}+4I_{xy}^{2}}\right]\\ \lambda_{2}=\frac{1}{2}\left[\left(I_{xx}+I_{yy}\right)-\sqrt{{\left(I_{xx}-I_{yy}\right)}^{2}+4I_{xy}^{2}}\right] \end{array}\right.$$
the eigenvectors corresponding to $\lambda_{1}$ and $\lambda_{2}$ are $e_{1}$ and $e_{2}$, which are obtained by the following formula: (19)$$e_{1}=\left(\begin{array}{c} 2I_{xy}\\ I_{yy}-I_{xx}+\sqrt{{\left(I_{xx}-I_{yy}\right)}^{2}+4I_{xy}^{2}} \end{array}\right)$$
(20)$$e_{2}=\left(\begin{array}{c} 2I_{xy}\\ I_{yy}-I_{xx}-\sqrt{{\left(I_{xx}-I_{yy}\right)}^{2}+4I_{xy}^{2}} \end{array}\right)$$

Obviously, through the observation of Formula (Equation 19), we can see that $\lambda_{1}\geq \lambda_{2}$. In reference [54], it is pointed out that because $\lambda_{1}\geq \lambda_{2}$, the feature vector $e_{1}$ has the largest second-order directional derivative direction in all directions, which is considered as the direction passing through the image feature, and $e_{2}$ is considered as the direction along the image feature. Using the eigenvalues and eigenvectors of the derived Hessian matrix, we construct the diffusion matrix D in Formula (Equation 16). The eigenvector of D is used as the eigenvector of H. We use $\eta_{1}$,$\eta_{2}$ to represent the two eigenvalues of D, as shown in the following formula: (21)$$\left\{\begin{array}{c} \eta_{1}=\frac{1}{1+{\left(|\nabla I|/K\right)}^{2}}\\ \eta_{2}=1 \end{array}\right.$$
where *K* serves as a threshold, and finally, the D takes the following form,
(22)$$D=\left(e_{1}e_{2}\right)\left(\begin{array}{cc} \eta_{1} & 0\\ 0 & \eta_{2} \end{array}\right){\left(e_{1}e_{2}\right)}^{T}$$

From Formula (Equation 22) we can get some information: (I) when $\left|\nabla I\right|\to \infty ,\;\eta_{1}\to 0$, the HBGVF snake will give up continuing to spread along the image gradient direction on the boundary and spread on the boundary. Therefore, the noise on the image edge can be eliminated while the image edge is preserved; (II) when $\left|\nabla I\right|\to 0,\;\eta_{1}\to 1=\eta_{2}$, that is, in the homogeneous region, the diffusion is isotropic, which is beneficial to the elimination of noise.

Through the above methods, the HBGVF model will have anisotropy, so it can accurately converge all kinds of concave surfaces and retain the weak edge of the image. The methods in reference [61] are used for reference to solve the model proposed in this paper, and the source code in Matlab is available to the public upon request. We note that, since the Hessian matrix and the diffusion matrix should be calculated, the computation time of the proposed HBGVF model is longer than the original GVF model.

## 4. Corresponding Comparative Experiments

In the experimental part, we show the important characteristics of the HBGVF model by comparing the HBGVF model with GVF [25], GGVF [57], VEF [58], NGVF [38], and CN-GGVF [41]. We normalized the image intensity to the [0,1] range, set and α, β to 0.1, and set the time step for all snakes with the size of τ=0.5. For an image of size M·N, the iteration for the calculation of all GVF-like models is $\sqrt{M\cdot N}$, and the time step is 1(less than 1/(4μ)). In order to get a large capture range, μ is 0.2 for GVF, NGVF, and HBGVF, *k* is 0.5 for the GGVF and CN-GGVF, the region D for the VEF model is of size M·N, *k* for the HBGVF is 0.1, unless otherwise stated.

### 4.1. Common Concerns for the GVF-Like Snakes

The GVF model was originally proposed to overcome the shortcomings of traditional gradient-based external force, such as narrow capture range and poor convergence on concave surfaces. Through the following experiments, we will prove some excellent characteristics of HBGVF snake compared with the GVF snake, such as large capture range, accurate convergence to concave, and insensitive to image initialization. Figure 1 shows the convergence results of HBGVF snake on a oom image, U-shaped image, and main body contour respectively. The gray dotted line is the initial contour, and the red solid line is the convergence result. It can be observed from the figure that the HBGVF snake converges to the U-shaped concave surface and is automatically connected to the subject contour. It can be seen from the initialization results in the figure that the HBGVF snake has the advantages of being insensitive to the initial contour and large capture range.

### 4.2. Convergence to Concavities

It can be seen from Figure 1 that the HBGVF snake performs well on converging to the U-shape image. Next, in order to better test the advantages of the HBGVF snake, we use the other three images with different concave surfaces to compare with other methods similar to GVF. Figure 2 presents the convergence results of the corresponding approaches. One can see that just the HBGVF and GGVF snakes can converge on the three images, the reason behind this observation is that the HBGVF model takes into account the image structure that was characterized by the Hessian matrix, and the GGVF model emphasizes the image structure by paying more attention to the edges by using two varying weighting functions. However, the CN-GGVF model also adopts the two varying weighting functions that are identical to those in the GGVF model, the CN-GGVF snake cannot converge to the various concavities at all, the reason is that the component normalization operation changes the direction of the vector field. Taking the heart image as an example, Figure 2i presents the associated GGVF vector field around the entrance of the concavity, one can see that the vector field in the blue circle is approximately horizontal, since the vector left to the blue circle is downward, it drives the snake contour into the concavity. Figure 2h presents the associated CN-GGVF vector field, where the vectors in black and red are these before and after component normalization, respectively, it is clear that the CN-GGVF field in the yellow circle before component normalization (in black) is similar to the GGVF vector, however, due to the component normalization, the CN-GGVF vector (in red) is upward, and pushes the snake contour out of the concavity. As a result, the CN-GGVF snake stops at the upper half of the concavity. This example tells us that component normalization is not always beneficial to the evolution of the snake contour. The concavities in the man and cat images are semi-close, and the CN-GGVF snake is also not good at converging to these concavities. Therefore, the improper use of the weighting function may cause the opposite effect. Of course, the appropriate use can greatly improve the accuracy of the model, such as the application in [62]. The GVF and VEF snakes just failed in one case, and we will see later that the shortcoming of the VEF snake is that it does not perform well when preserving weak edges. The NGVF snake just works well on the man concavities, and since the limited capture range, the initial contour for the cat image is very close to the cat at the left-bottom corner.

### 4.3. Weak Edge Preserving

Figure 3a is an example of testing the ability of the HBGVF model to retain the weak edge of the image. The outer ring of the image is seriously blurred in the upper right corner. Refer to the edge diagram in Figure 3b. It can be seen that the contour of the snake is easily attracted to the inner ring of the strong edge. Since it is a pair of contradictions to enlarge capture range and to preserve weak edge simultaneously, the regularization parameters are tuned to μ is 0.1 for GVF, NGVF, and HBGVF, *k* is 0.01 for the GGVF and CN-GGVF, the size of region D for VEF model is just one twenty-fifth of that of the image, *k* for HBGVF is 0.01. One can see that the HBGVF snakes can preserve the weak edge well although the diffusion parameter μ is identical to those of the GVF and NGVF; the reason behind this observation is that the HBGVF model takes into account the image structure. Although the kernel size for the VEF is very small and the initial contour for the VEF snake is close to the object, the snake contour yet collapsed at the weak edge. The CN-GGVF and GGVF snakes also stop at the weak edge and the convergence results are almost identical due to the similar diffusion mechanism, when compared with that of the HBGVF snake, the result of the HBGVF snake is smoother, this observation implies the HBGVF field is more regular.

### 4.4. Test Results of HBGVF Model on Real Images

In order to further highlight the comprehensive performance of the HBGVF snake, we used several real images for comparison. Figure 4 presents a gear image, where there are more than ten semi-close concavities with order number.

The parameter *k* is 0.2 for the GGVF and 0.3 for the CN-GGVF in order to get a balance between entering the concavities and preserving a weak edge, the parameters for other models are identical to those in Figure 1 and Figure 2. One can see that the GVF snake converges to the concavities from #0 to #9, although it collapses at the two teeth around concavity 5. The GGVF snake converges to the concavities from #0 to #8, and it seems that the GGVF snake is good at preserving a weak edge, in fact, one can see that there is contour entanglement from the right part in Figure 4d, which is a zoomed-in version of the blue rectangle in the left part. The VEF snake suffers from weak edge leakage, and collapses at most of the teeth, see Figure 4e. Figure 4f is the result of the NGVF snake treatment, which manifests that the NGVF snake is not good at concave convergence, and this observation agrees with that in Figure 2. The CN-GGVF snake performs similarly to the NGVF snake, see Figure 4g. Since the HBGVF takes into account the image structure, the HBGVF snake converges to the concavities from #0 to #12 except the 11th one. However, from the zoomed-in part of the blue rectangle in the left part, HBGVF snakes also performed poorly, as shown in the right part of Figure 4h; in fact, the performance in this example can be enhanced by decreasing the parameter *k* in HBGVF.

Figure 5 presents a second real image, a flying eagle, and the feathers on the wings are difficult for the active contour to extract. In order to get a balance between extracting the feathers on the wings and enlarging the capture range, the regularization parameter μ is 0.05 for GVF, NGVF, and HBGVF, *k* is 0.05 for the GGVF and CN-GGVF, the size of region D for the VEF model is just one sixty-forth of that of the image, *k* for HBGVF is 0.01. The white dash-point lines are the initial contour and the red solid lines are the convergent results. As can be seen from Figure 5a, the GVF snake works well except for the feathers on the right wing. Figure 5b shows that the GGVF snake yields good results in extracting the feathers on both wings, however, it is trapped in local minimum behind the tail. The result of the VEF snake is reported in Figure 5c, and it is obvious from the results that the snake contour is trapped in a local minimum and also fails on extracting the feathers. The NGVF and CN-GGVF snakes are also trapped in a local minimum, see Figure 5d,e, respectively, and the CN-GGVF snake cannot enter the concavities formed by the feathers. On the contrary, Figure 5f shows that the HBGVF snake works well on extracting the feathers and is not trapped in a local minimum, which manifests that the HBGVF field is regular.

Figure 6 presents a medical image, and for the weak edge shown in the white box in Figure 6a, the snake contour is prone to leakage here, the intensity inhomogeneity is also a difficulty. In order to achieve a balance between maintaining the weak edge and overcoming inhomogeneity, the regularization parameter μ is 0.02 for GVF, 0.03 for NGVF and HBGVF, *k* is 0.03 for the GGVF, and 0.07 for the CN-GGVF, the size of region D for VEF model is just 1/144 of that of the image, *k* for HBGVF is 0.01. One can see that there is weak-edge leakage and local minimum trap simultaneously for the GVF, VEF, and NGVF snakes. The GGVF, CN-GGVF, and HBGVF snakes yield similar results, where there is no weak-edge leakage or local minimum trap. It is clear that the μ for HBGVF and NGVF are identical, and even larger than that for the GVF, however, the HBGVF snake preserves a weak edge well, the reason behind this observation is that the HBGVF model takes into account the image structure. Figure 7 shows more results of the HBGVF snake on real images, the initial contours are dash-point lines and the convergence results are the solid red lines. The first row presents flowers and leaves and the HBGVF snake extracts the objects accurately, the second row shows three eagles and the difficulty for the HBGVF snake is similar to that in Figure 5, the HBGVF snake also yields satisfactory results. There are three medical images in the third row, and in each panel, the image on the left is the original image with initial contour, from which one can see the blurred and weak boundaries of the objects. The display result on the right shows that the HBGVF snake can satisfactorily delineate the object boundaries.

## 5. Conclusions

To sum up, the smoothness constraint formula is expressed in the form of a matrix, and the image structure represented by the Hessian matrix is introduced into the GVF model. This GVF model based on the Hessian matrix is abbreviated as HBGVF. Through the above theoretical analysis and experimental comparison, it can be proved that compared with other GVF-based models, the HBGVF snake has many advantages, such as excellent convergence on various concave surfaces, retaining weak edges, and so on. The above experiments include synthetic images and real images in real life. These experiments have proved the excellent characteristics of the HBGVF model. The proposed HBGVF model can also be employed for other applications such as those in [63,64,65,66,67,68,69,70,71,72], and this is our next goal. 

## Figures and Tables

**Figure 1 sensors-22-04956-f001:**
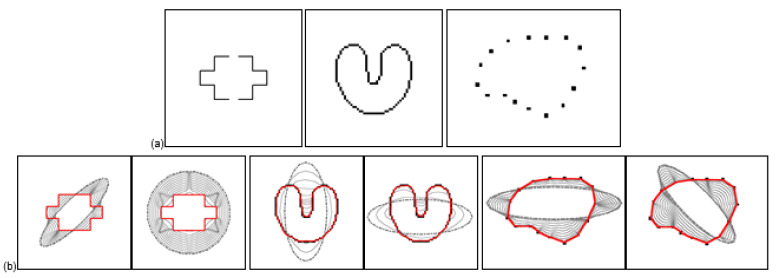
(**a**) Test images: room image, U image, and subject contour. (**b**) Convergence results with different initializations and evolutions of the HBGVF snakes.

**Figure 2 sensors-22-04956-f002:**
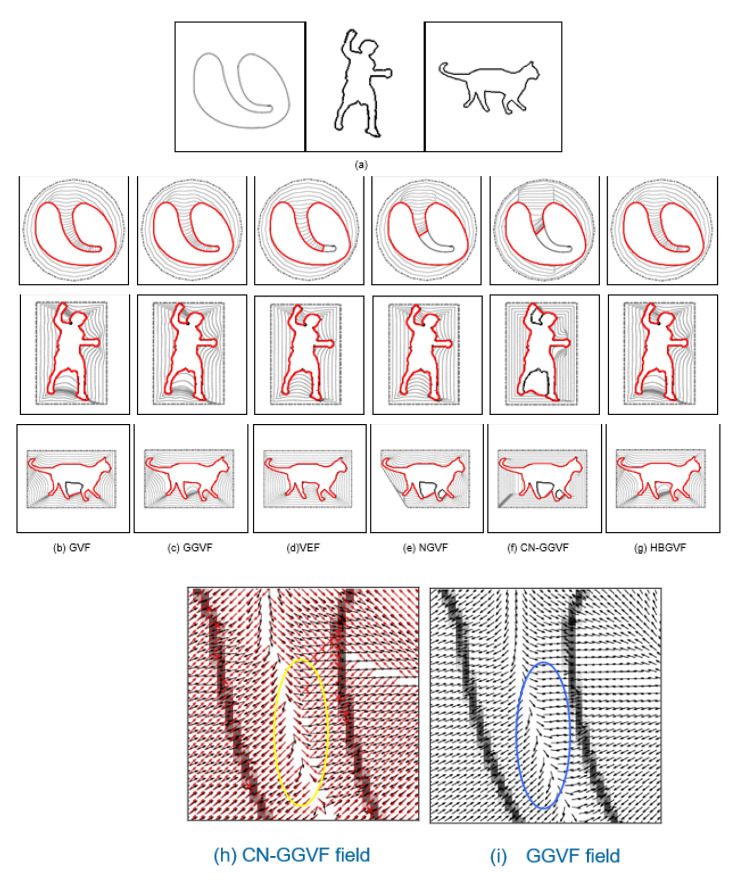
Convergence to concavities. (**a**) Test images: heart image, man image, and cat image. Evolution and convergence results of the (**b**) GVF snake, (**c**) GGVF snake, (**d**) VEF snake, (**e**) NGVF snake, (**f**) CN-GGVF snake, and (**g**) HBGVF snake. (**h**) The CN-GGVF field, the vectors in black and red are these before and after component-normalization, respectively, (**i**) the GGVF field.

**Figure 3 sensors-22-04956-f003:**
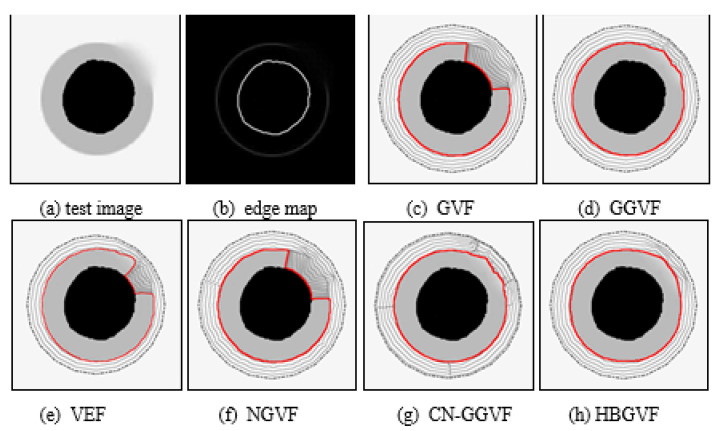
(**a**) Test image, (**b**) edge map. Convergence results of each model: (**c**) the GVF snake, (**d**) the GGVF snake, (**e**) the VEF snake, (**f**) the NGVF snake, (**g**) the CN-GGVF snake, and (**h**) the HBGVF snake.

**Figure 4 sensors-22-04956-f004:**
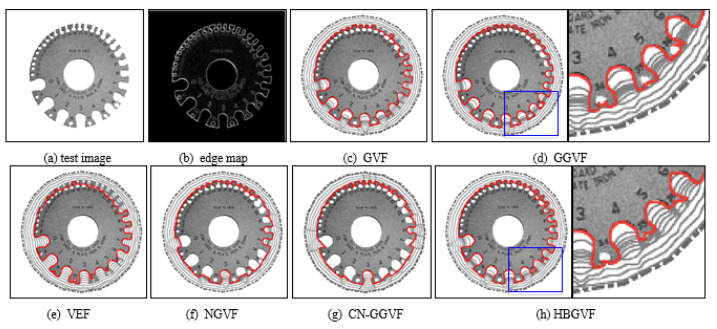
(**a**) Original test metal gauge image; (**b**) edge map; the convergence results of each model: (**c**) the GVF snake, (**d**) the GGVF snake, (**e**) the VEF snake, (**f**) the NGVF snake, (**g**) the CN-GGVF snake, and (**h**) the HBGVF snake.

**Figure 5 sensors-22-04956-f005:**
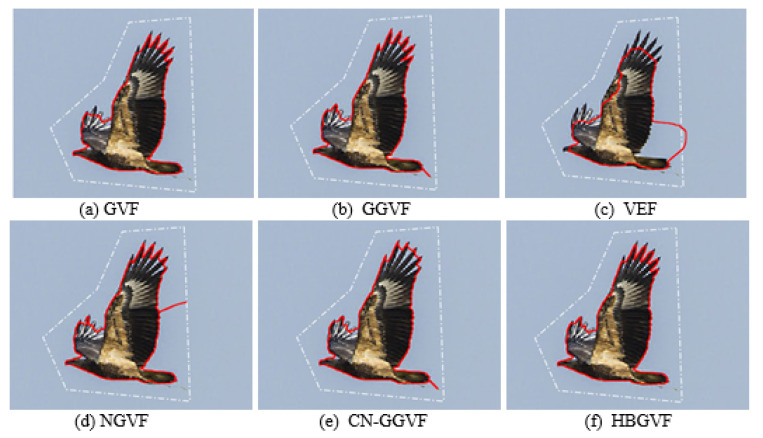
The convergence results of each model: (**a**) the GVF snake, (**b**) the GGVF snake, (**c**) the VEF snake, (**d**) the NGVF snake, (**e**) the CN-GGVF snake, and (**f**) the HBGVF snake. In order to get a balance between preserving the feathers on the wings and enlarging the capture range, the regularization parameter μ is 0.05 for GVF, NGVF, and HBGVF, *k* is 0.05 for the GGVF and CN-GGVF, the size of region D for VEF model is just one sixty-forth of that of the image, *k* for HBGVF is 0.01.

**Figure 6 sensors-22-04956-f006:**
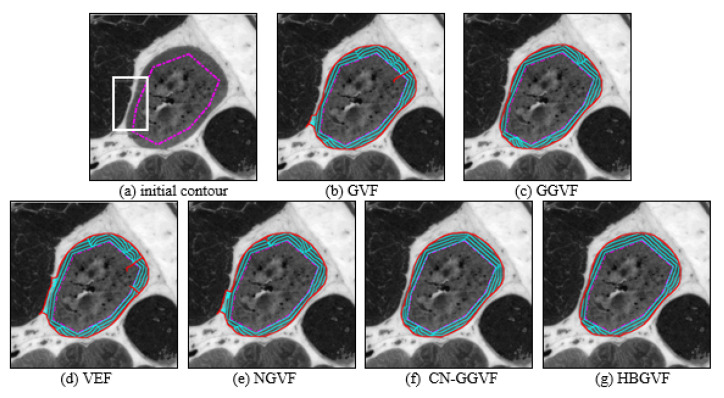
(**a**) Test medical image; the convergence results of each model: (**b**) the GVF snake, (**c**) the GGVF snake, (**d**) the VEF snake, (**e**) the NGVF snake, (**f**) the CN-GGVF snake, and (**g**) the HBGVF snake.

**Figure 7 sensors-22-04956-f007:**
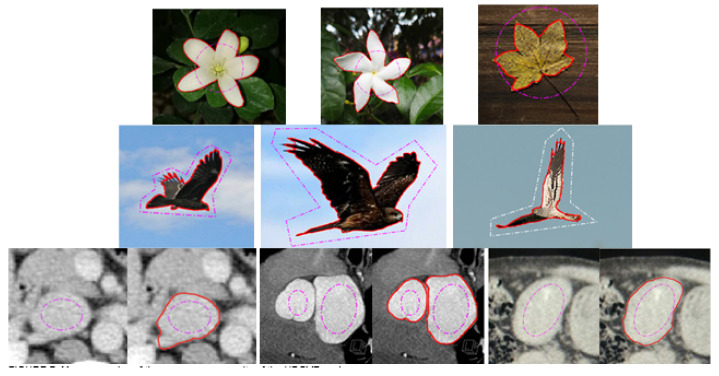
More examples of the convergence results of the HBGVF snake.

## Data Availability

Dateset is available at request.

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
