# Peer review of "Image Segmentation Using Active Contours with Hessian-Based Gradient Vector Flow External Force"

_sensors, 2022, doi:10.3390/s22134956_

Round 1
Reviewer 1 Report
This paper presents the new image segmentation method using the hessian-based gradient vector flow external force. I think this paper is suitable to publish but it has some points to improve before accepting this paper:
1) I think the results of this work are enough to explain your method is the best method. However, I think this paper must be shown the metrics to show more performance of your algorithm. For example, the computation time or the error of every method compared with the true results in percent, etc. I think it will be shown the advantage of your proposed method clearly.
2) Why the authors haven't compared the results with the deep learning results? I found one famous paper about this method. "Minaee, Shervin, et al. "Image segmentation using deep learning: A survey." IEEE transactions on pattern analysis and machine intelligence (2021)."
3) Is it possible to do more test image and show the results only the original image and the metric.
Author Response
Dear Professor,we would like to thank you for your hard work for our manuscript and put forward suggestions on how to improve our paper, from which we have learned a lot! We also apologize for the delay in revising the manuscript.
According to the reviewer's suggestions, we revised the paper to make the manuscript more readable. We also examined the English version of the manuscript carefully. In addition, a point by point response to reviewers' comments is attached below.
We hope that these changes will lead to the acceptance of the manuscript and look forward to your reply!
Best Regards with many thanks!
Yours Sincerely,
Yuanquan Wang, etc
Reviewer #1: This paper presents the new image segmentation method using the hessian-based gradient vector flow external force. I think this paper is suitable to publish but it has some points to improve before accepting this paper:
Response: Thank you very much for your nice comments on the paper! A point-by-point answer to your comments is as follows:
- I think the results of this work are enough to explain your method is the best method. However, I think this paper must be shown the metrics to show more performance of your algorithm. For example, the computation time or the error of every method compared with the true results in percent, etc. I think it will be shown the advantage of your proposed method clearly.
Answer: Thank you for your very good suggestion. It is really a very important issue to show some metrics on the algorithm. The motivation of the HBGVF model is to enhance the performance of the GVF model on concavity convergence and weak edge preservation, so we present various cases on these two issues, and visual inspection is employed to evaluate the results. On the other hand, the computation time is slightly longer than the original GVF model, and we have added some comments on this point in the revised manuscript.
Again with thanks for your comments, and we will keep your comments in mind and put them into action in future.
- Why the authors haven't compared the results with the deep learning results? I found one famous paper about this method. "Minaee, Shervin, et al. "Image segmentation using deep learning: A survey." IEEE transactions on pattern analysis and machine intelligence (2021)."
Answer: Thank you for your suggestions! Yes, deep learning has achieved great success in many image-based applications, including segmentation. However, the deep learning methods need big data for training, and the motivation of the HBGVF model is to enhance the performance of the GVF model on concavity convergence and weak edge preservation, the deep learning and HBGVF belong to different categorical methods for image segmentation.
Many thanks for pointing out this reference, and we have added it to the ref. list, and more works on deep learning are mentioned in the introduction section.
3) Is it possible to do more test image and show the results only the original image and the metric.
Answer: Thank you for your suggestions!
At present, the images in the experiments are employed to validate the motivation of the HBGVF model, i.e., to enhance the performance of the GVF model on concavity convergence and weak edge preservation, one can see the enhanced performance of the HBGVF from these images.
Anyway, your suggestions are very good!
Reviewer 2 Report
This article proposed a HBGV for image segmentation. The HBGVF model based on hessian has many advantages over other relevant GVF methods, such as accurate convergence to various concave surfaces, excellent weak edge retention ability and so on.
The idea is sound, but the following comments must be considered.
- Image segmentation is also widely studied in more other fields, and the authors should consider in the introduction and so on, e.g., hog-shipclsnet that used the edge features similar to the practice of the authors, quad-fpn, ls-ssdd, hyperli-net, balance scene learning mechanism, shipdenet-20, squeeze-and-excitation laplacian pyramid network with dual-polarization feature fusion, sar ship detection dataset (ssdd).
- Line 2, “further results” -> “better results”;
- Line 6, “The GVF” (HBGVF)?
- Line 12, add some work about deep learning methods, e.g., full-level context squeeze-and-excitation roi extractor.
- Line 68, avoid [3] in the section title, and other places in the full texts.
- In fact, detection, classification, and segmentation share many similar techniques, e.g., polarization fusion network with geometric feature embedding, balance learning for ship detection from synthetic aperture radar remote sensing imagery.
- The authors should compare these sota methods, or they maybe declare them in some suitable places.
- It will be welcome that a full-level context squeeze-and-excitation roi extractor for sar ship instance segmentation, and htc+ for sar ship instance segmentation, can be added and considered. They belong to image segmentation.
- Reconsider after major revision (control missing in some experiments)
Author Response
Dear Professor,
we would like to thank you for your hard work for our manuscript and put forward suggestions on how to improve our paper, from which we have learned a lot! We also apologize for the delay in revising the manuscript.
According to the reviewer's suggestions, we revised the paper to make the manuscript more readable. We also examined the English version of the manuscript carefully. In addition, a point by point response to reviewers' comments is attached below.
We hope that these changes will lead to the acceptance of the manuscript and look forward to your reply!
Best Regards with many thanks!
Yours Sincerely,
Yuanquan Wang, etc
Reviewer #2: This article proposed a HBGV for image segmentation. The HBGVF model based on hessian has many advantages over other relevant GVF methods, such as accurate convergence to various concave surfaces, excellent weak edge retention ability and so on. The idea is sound, but the following comments must be considered.
Response: Thank you very much for your nice comments on the paper! A point-by-point answer to your comments is as follows:
Image segmentation is also widely studied in more other fields, and the authors should consider in the introduction and so on, e.g., hog-shipclsnet that used the edge features similar to the practice of the authors, quad-fpn, ls-ssdd, hyperli-net,balance scene learning mechanism, shipdenet-20, squeeze-and-excitation laplacian pyramid network with dual-polarization feature fusion, sar ship detection dataset (ssdd).
Answer: Thank you for pointing out these important and interesting works!
We have added them to the ref list and commented them in the introduction section. We have learnt much from these thought-provoking works!
Again with thanks!
Line 2, “further results” -> “better results”;
Line 6, “The GVF” (HBGVF)?
Line 68, avoid [3] in the section title, and other places in the full texts.
Answer: Thanks a lot! We have corrected these typos.
Line 12, add some work about deep learning methods, e.g., full-level context squeeze-and-excitation roi extractor. In fact, detection, classification, and segmentation share many similar techniques, e.g., polarization fusion network with geometric feature embedding, balance learning for ship detection from synthetic aperture radar remote sensing imagery. The authors should compare these sota methods, or they maybe declare them in some suitable places.It will be welcome that a full-level context squeeze-and-excitation roi extractor for sar ship instance segmentation, and htc+ for sar ship instance segmentation, can be added and considered. They belong to image segmentation.
Answer: Many thanks for these suggestions!
Deep learning really plays important role in many image-based applications, such as detection, classification, and segmentation, and many excellent works have been proposed, such as queeze-and-excitation roi extractor, and etc.
We have studied them and gave some comments on these works in the introduction section, and finally, they are in the ref. list!
Reconsider after major revision.
Answer: Many thanks for your encouragement, and we hope the current manuscript would meet with your approval and will be accepted soon!
Round 2
Reviewer 1 Report
The reviewer can answer all of my questions. However, the authors must check the format of your paper for example Eq.(7).
Author Response
Thank you very much for the hard work of the two reviewers on our manuscript. Now the peer-to-peer replies to the reviewers are as follows:
(1) Modify the manuscript according to the review comments. It is suggested to check the format of your paper, such as formula (7). We have checked the format of the manuscript, including formula (7) and some references.
Reviewer 2 Report
Accept. No more comments.
Author Response
Thank you very much for the hard work of the editor and the two reviewers on our manuscript. Now the point-to-point replies to the editor and reviewers are as follows.